# Effects of Virtual Reality (VR)-Based Meditation on Sleep Quality, Stress, and Autonomic Nervous System Balance in Nursing Students

**DOI:** 10.3390/healthcare12161581

**Published:** 2024-08-08

**Authors:** Ki-Yong Kim, Myung-Haeng Hur, Won-Jong Kim

**Affiliations:** 1Department of Nursing, Gimcheon University, Gimcheon-si 39528, Gyeongsangbuk-do, Republic of Korea; kykim0215@gimcheon.ac.kr; 2Department of Nursing, Eulji University, Uijeongbu-si 11759, Gyeonggi-do, Republic of Korea; mhhur@eulji.ac.kr

**Keywords:** autonomic nervous system, meditation, nursing student, sleep quality, stress, virtual reality

## Abstract

This study aimed to determine the effects of virtual reality (VR)-based meditation on the sleep quality, stress, and autonomic nervous system balance of nursing students. Nursing students were categorized into experimental groups I (VR-based meditation) and II (concentration meditation) and a control group. Before the study’s initiation, we measured the participants’ general characteristics, and a sleep measurement device was used to measure sleep quality. Stress levels and heart rate variability were measured before and after study completion. On the first day, all three groups slept without any intervention. On days 2–6, only experimental groups I and II implemented the intervention before sleeping. We found that the subjective sleep quality, wakefulness after sleep onset, sleep efficiency, deep sleep quality, subjective stress, objective stress, and autonomic nervous system balance of the VR meditation group were significantly better than those of the other groups. Our results reveal that the participants who underwent VR-based meditation experienced better sleep quality, lower stress levels, and improved autonomic nervous system balance compared with those in the concentration meditation and control groups. Thus, VR-based meditation effectively enhances sleep quality, lowers stress levels, and improves autonomic nervous system balance in nursing students.

## 1. Introduction

The academic burden, interpersonal relationships, group activities, and economic burden of students change once they enter universities. Accordingly, they often experience psychological difficulties and sleep disorders [1,2]. Particularly, nursing students experience high stress levels as part of their theoretical learning, clinical practice, and professional responsibility [3,4]. Continuous exposure to stressors can negatively affect overall health, cause fatigue and depression, and reduce sleep quality [5,6,7].

Sleep is an essential physiological process for life in human beings [8]. Appropriate sleep quality facilitates effective cellular recovery and growth, enhances psychological stability, and reduces stress [9,10]. Conversely, long-term persistence of low sleep quality results in impaired learning ability, stress, hypertension, and cardiac disease, increases accident risk, and affects the quality of life [9,11,12,13].

Nursing students are continually exposed to complex stress factors, such as those associated with theoretical learning, clinical practice, and interpersonal relationships, and continual, diverse stresses that reportedly cause autonomic nervous system (ANS) imbalance [14,15,16]. The ANS imbalance resulting from long-term stress ultimately causes various physical and psychological issues, such as fatigue, sleep problems, and anxiety [9]. Therefore, maintaining ANS balance is important. Additionally, ANS stabilization directly improves stress management, stabilizes heart rate and blood pressure, improves sleep quality, enhances physical rest, and hastens fatigue recovery [17].

Scholars propose various interventions to alleviate stress and correct ANS imbalance. Among them, meditation is a cost-effective and objective method to enhance psychological stability [18]. It is used by many as a therapeutic method for stress management. Further, any improvement in learning ability, psychological stability, and sleep quality reportedly stabilizes brain activity [19,20,21,22]. With the advancement of technology, the use of virtual reality (VR) is increasing in various fields. Currently, VR is used in the field of integrated meditation to promote psychological stability, improve sleep quality, overcome time and location constraints to increase convenience, increase immersion, minimize environmental interference, and promote deep concentration [23,24,25,26]. Although several interventional studies have examined meditation [27] and mindfulness meditation [28,29], not many have combined meditation and VR [30,31,32].

Furthermore, to date, there are currently not many intervention studies examining how VR-based meditation affects sleep quality, stress levels, and ANS balance. Accordingly, in this study, we aimed to subject eligible nursing students to VR-based meditation and report its effects on sleep quality, stress, and ANS balance. In this manner, we aimed to provide basic data to help construct an educational environment that facilitates the development of specific, personalized programs to nurture positive lifestyle habits in nursing students and enable the recovery of these students’ psychological and physical health.

**Hypothesis** **1.**
*Applying virtual reality-based meditation will affect sleep quality.*


**Hypothesis** **2.**
*Applying virtual reality-based meditation will affect stress.*


**Hypothesis** **3.**
*Applying virtual reality-based meditation will affect ANS balance.*


## 2. Materials and Methods

### 2.1. Research Design

This study was conducted as a randomized controlled experiment with three groups of participants: control group (no intervention), experimental group I (VR-based meditation), and experimental group II (concentration meditation) (Figure 1).

### 2.2. Participant Selection and Sample Size

#### 2.2.1. Participant Selection

Nursing students studying in U City, D Metropolitan City, and G Province and who met the selection criteria participated in this study. Before data collection, this study was approved by the Institutional Review Board (EU23-27) and the Clinical Research Information Service (KCT0009016). Recruitment notices were then posted on social networks and department notice boards. Table 1 depicts participants’ inclusion and exclusion criteria.

#### 2.2.2. Sample Size

The sample size was calculated using G*Power 3.1.9.7 (University of Düsseldorf, Düsseldorf, Germany). The number of participants in this study was calculated using the F-test (analysis of variance [ANOVA]) with an effect size of 0.25, a significance level of 0.05, and a power of 0.95. Therefore, the required sample size was 18 per group, for a total of 54. Based on the participants’ characteristics (age, school year, area of residence, etc.) and the application of VR-based meditation, we recruited 60 participants (20 per group) after considering an expected dropout rate of 10%.

### 2.3. Randomization and Blinding

Initially, 70 participants expressed their desire to participate in this study. Among them, 10 individuals who had sleep quality scores above the cutoff value of 66 points were excluded from this study. The remaining 60 participants were allocated to investigational group I, investigational group II, and the control group: 20 participants per group (Figure 2), by block blind randomization using MS Excel’s random number generator function. No dropouts occurred during data collection, and the data of all 60 participants were used in the final analysis.

### 2.4. Data Collection

Data were collected between 12 September and 17 October 2023. Participants were recruited between 12 and 17 September 2023. The subjective data of each variable were measured through a questionnaire. For objective data measurement of sleep quality, an activity tracker tool in the form of a wristwatch was used. Stress and HRV were measured at the individual’s index finger in the form of a pulse oximeter. Data were first collected from the control group to prevent any diffusion or contamination during the experimental procedure.

### 2.5. Experimental Treatment and Procedure

#### 2.5.1. Experimental Treatment

VR-Based Meditation. VR-based meditation involves the application of traditional meditation techniques in specific environments or situations that are similar to reality but are artificially generated. Using a head-mounted display (Oculus MetaQuest 2, Meta Platforms, Inc., Menlo Park, CA, USA) and utilizing the processing of visual and auditory information in the brain, we presented lifelike virtual worlds in the form of immersive and realistic 360°, 4K videos, which could be viewed from users’ perspectives, to provide psychological stability to participants. Four video types (sea, night sky, walking through a forest in the fall, and walking through a green forest) were generated. The participants selected a video of their preference for meditation.

Concentration Meditation. A focused mind is tranquil and concentrates on a specific object/process, such as breathing. Therefore, concentration meditation is a type of meditation in which the participants continually focus on a single object, and their minds naturally become tranquil over time [25]. In this study, the participants performed concentration meditation in a relaxed state under minimal environmental noise.

Measurement environment. The measurement site area is 19.83 m^2^, and the indoor temperature is set to 25 °C considering the temperature that makes the subject feel comfortable [33]. The laboratory has a window for good ventilation, and a sofa, table, and chair are placed to provide a comfortable environment for the subject.

#### 2.5.2. Experimental Procedure

Experimental Group I.

For pre-intervention measurements, each participant was provided with a VR device and a sleep-monitoring device. The VR device was preinstalled with the VR-based meditation application used in this study. The participants were provided instructions to use the application, and their ability to do so was directly verified. The device was confirmed to operate in connection with a sleep-monitoring activity tracker (Fitbit Charge 4, FitBit^®^, Inc., San Francisco, CA, USA) and a personal mobile phone. To apply the correct breathing method, a meditation instructor was invited, training was provided on the breathing method, and leaflets on the method were provided. On day 1, the participants went about their daily lives without any intervention for the pretest measurement. To encourage application of the program, text messages were sent at a specific time, and responses were checked. Starting from the second day, a VR-based meditation program was applied for 30 min each day for 5 days between 10 PM and midnight [34]. Before applying VR-based meditation, the breathing method learned in advance was applied to stabilize breathing, and then VR-based meditation was applied. Post-measurements (sleep quality, stress, ANS balance) were performed on day 7.

Experimental Group II.

For pre-intervention measurements, participants were provided with a sleep-monitoring device and instructions. The device was confirmed to operate in connection with a sleep-monitoring activity tracker and a personal mobile phone. To apply concentration meditation, a meditation instructor was invited, training was provided on breathing techniques and concentration meditation, and leaflets on the method were provided. On day 1, the participants went about their daily lives without any intervention for pretest measurement. On day 1, the participants went about their daily lives without any intervention for the pretest measurement. To encourage application of the program, text messages were sent at a specific time and responses were checked. Starting from the second day, a concentration meditation program was applied for 30 min each day for 5 days between 10 PM and midnight [34]. The participants performed concentration meditation before going to sleep. Before starting to meditate, they adopted a comfortable position and closed their eyes. Subsequently, they inhaled deeply and slowly, held their breath for a moment, and then exhaled slowly. After stabilizing their breathing, participants performed concentration meditation. On losing their focus during meditation, the participants repeated the process mentioned above until they had performed concentration meditation for 30 min. Post-measurements (sleep quality, stress, and ANS balance) were performed on day 7.

Control Group.

For pre-intervention measurements, participants were provided with a sleep-monitoring device. The device was confirmed to operate in connection with a sleep-monitoring activity tracker and a personal mobile phone. During the experiment, participants continued their daily lives without any intervention. Post-measurements (sleep quality, stress, and ANS balance) were performed on day 7.

### 2.6. Research Tools

#### 2.6.1. Sleep Quality

Subjective Sleep Quality. Subjective sleep quality was measured using the Korean Modified Leeds Evaluation Questionnaire (KMLSEQ) adapted by [35] from the Leeds Sleep Evaluation Questionnaire (LSEQ), which was originally developed by Parrott and Hindmarch [36]. The modified questionnaire consists of 10 questions, and the total score ranges from 0 to 100 points, with higher scores indicating better sleep quality. Any score below the cutoff value of 66 points indicates poor sleep quality.

In terms of reliability, LSEQ and KMLSEQ revealed Cronbach’s α values of 0.92 and 0.95, respectively, at the time of development. In this study, the instrument had a Cronbach’s α value of 0.950.

Objective Sleep Quality. Objective sleep quality was measured using an activity tracker (Fitbit Charge 4, FitBit^Ⓡ^, Inc., San Francisco, CA, USA) that detected physical movements and heart rate variability (HRV) by using a three-dimensional micro-electromechanical system and photoplethysmography. In this study, objective sleep quality was measured in terms of the following factors:

Wakefulness after sleep onset (WASO): The total time (in minutes) that a participant was awake between falling asleep and waking up.

Sleep efficiency (SE): The percentage of time spent sleeping, excluding WASO, by a participant.

Deep sleep time (DST): The percentage of time recorded by FitBit Charge 4 as deep sleep duration, which corresponds to non-rapid eye movement (NREM) stage 3 sleep.

#### 2.6.2. Stress

Subjective Stress. Subjective stress was measured using the Korean Perceived Stress Scale (K-PSS), which was adapted and validated by Lee et al. [37] from the Perceived Stress Scale originally developed by Cohen et al. [8]. This instrument is measured on a 5-point Likert scale, in which higher scores indicate higher stress. The Cronbach’s α values were 0.82 in the study by Lee et al. [37] and 0.804 in the current study.

Objective Stress. Stress index is a value that quantifies stress using a standard induction method. In this method, a pulse oximeter (Canopy 9 RSA, IEMBIO Co., Ltd., Gangwon-do, Republic of Korea) is placed on an individual’s index finger to measure HRV and ANS balance for 5 min prior to analysis. The stress index ranges from 1 to 10, with higher numbers indicating higher levels of exposure to stressful situations.

#### 2.6.3. Autonomic Nervous System Balance

Autonomic Nervous System Balance. Participants’ ANS balance was analyzed using the HRV index, which is a standard derivation method based on analyzing the heart rate gradients measured for 5 min by placing a pulse oximeter on the index finger. The HRV index ranges from 7.0 to 12.0, with higher values indicating better maintenance of ANS balance [38].

### 2.7. Data Analysis

The collected data were analyzed using IBM SPSS Statistics 27.0 (IBM Corp., Armonk, NY, USA). Data analysis comprised the following steps:Participants’ general characteristics were analyzed by frequency, error, percentage, mean, and standard deviation; homogeneity was analyzed using a χ^2^-test and a one-way ANOVA.The homogeneity of the pre-intervention dependent variables of experimental group I, experimental group II, and the control group was analyzed using one-way ANOVA.A one-way ANOVA was used to test differences in sleep quality, stress, and ANS balance between the VR-based meditation, concentration meditation, and control groups. Further, Duncan’s test was used post hoc to test significant results. We used a repeated-measures ANOVA to verify the difference in effects caused by the change from pre- to post-intervention.

### 2.8. Ethical Considerations

The Institutional Review Board of Uijeongbu E University (EU23-27) approved this study. Before experimenting, the purpose and procedures of this study were explained to the participants, and they were informed that they could voluntarily opt out at any time. All participants voluntarily provided written informed consent for participation. The collected data were processed by assigning a unique identification number in accordance with personal information processing guidelines to protect participants’ personal information. Following the completion of data analysis, it was stored in a computer with a lock. The collected data will be stored for 3 years after the end of this study and then destroyed. Participants were provided with transportation expenses and a small gift following the completion of data collection. After appropriate measurements had been made, participants in experimental group II and the control group were given the opportunity to participate in a VR-based meditation program if they desired.

## 3. Results

### 3.1. Homogeneity Testing of Participants’ Pre-Intervention Characteristics and Dependent Variables

This study involved 60 participants, with 20 participants allocated to VR-based meditation, concentration meditation, and control groups. We used a χ^2^-test and a one-way ANOVA to test the homogeneity of the general characteristics of the three groups. Our analysis revealed no significant differences in age, sex, sleep duration, meditation, underlying disease, or vital signs; accordingly, the three groups were determined to be homogeneous (Table 2).

We used a one-way ANOVA to test the homogeneity of dependent variables. The test revealed no significant differences in sleep quality (subjective or objective), stress (subjective or objective), and ANS balance among the three groups; accordingly, the groups were considered homogeneous (Table 2).

### 3.2. Effects of Virtual Reality-Based Meditation on Sleep Quality

As a result of measurement after applying the intervention, there was a significant difference in subjective quality of sleep among the three groups (*p* < 0.001). In the post hoc test, experimental group I revealed a significant difference in subjective quality of sleep compared to the control group (*p* < 0.001) (Table 3).

Objective quality of sleep was confirmed by WASO, SE, and DST. As a result of measurement after applying the intervention, there was a significant difference in WASO among the three groups (*p* < 0.001). In the post hoc test, the WASO score of experimental group I was significantly different from that of the control group (*p* = 0.034). No significant differences occurred over time in the repeated-measures ANOVA. However, there were significant differences depending on the group *(p* < 0.001). Further, no significant interaction between time and group was observed.

There were significant differences in SE among the three groups (*p* < 0.001). In post hoc testing, experimental groups I and II showed significant differences in SE compared with the control group (*p* = 0.004). We performed a repeated-measures ANOVA to examine the impact of one week of VR-based meditation on SE. No significant difference was noted over time; however, significant differences among groups (*p* < 0.001) were recorded. A significant interaction effect between time and group (*p* = 0.018) was indicated.

There were significant differences among the three groups (*p* < 0.001). In post hoc testing, experimental group I showed significant differences in DST compared with experimental group II and the control group (*p* < 0.001). Repeated-measures ANOVA revealed significant differences by time (*p* < 0.001) and group (*p* < 0.001), as well. There was a significant interaction effect between time and group (*p* = 0.002), as well (Table 4).

### 3.3. Effects of Virtual Reality-Based Meditation on Stress

As a result of measurement after applying the intervention, there was a significant difference in subjective stress among the three groups (*p* = 0.015). In post hoc testing, experimental group I showed a significant difference in subjective stress score compared with the control group (*p* = 0.035). Moreover, it also showed significant differences in objective stress among the three groups (*p* < 0.001). In post hoc testing, experimental group I significantly differed in objective stress score from experimental group II and the control group (*p* < 0.001) (Table 5).

### 3.4. Effects of Virtual Reality-Based Meditation on Autonomic Nervous System Balance

As a result of measurement after applying the intervention, there were significant differences among the three groups (*p* = 0.044). In post hoc testing, the experimental group I showed more significant differences in ANS balance than the experimental group II and the control group (*p* < 0.001) (Table 5).

## 4. Discussion

The results of this study showed virtual reality-based meditation had a positive effect on sleep quality, stress, and ANS balance in nursing students.

This intervention was found to be an effective intervention with high practical utility in improving sleep quality, reducing stress, and stabilizing the autonomic nervous system of nursing students in their early adult years. In this study, subjective sleep quality had a significant effect on the virtual reality-based meditation group. In particular, objective sleep quality was significantly improved in WASO (minutes), SE (%), and DST (%) measured using an activity tracker. These results show that, compared to previous studies [39,40,41,42], which applied meditation to general adults and older people to determine subjective sleep quality, a virtual reality-based meditation intervention providing immersive real-life natural images improves concentration and induces psychological stability, in turn helping participants. In addition, this study is meaningful as it overcomes the limitations of previous studies, which focused on subjective sleep measurement, by measuring objective sleep quality.

This study’s virtual reality-based meditation intervention for increasing immersion and concentration was similar to the results of improved WASO, DST, and SE from a previous study that applied virtual reality meditation to intensive care unit patients [32]. In this study, virtual reality-based meditation, which provides actual natural images, was applied to improve concentration and induce psychological stability. The application of virtual reality-based meditation resulted in a positive effect of the intervention. It promoted voluntary health practices by providing autonomous motivation for health care related to sleep hygiene to nursing students in early adulthood [43].

Based on the results of this study, wakefulness after sleep onset (WASO, min) decreased in the experimental group. Additionally, deep sleep (DST, %) and sleep efficiency (SE, %) increased. The significant results of this study pertaining to WASO, SE, and DST indicate the possibility of effective sleep improvement through VR-based meditation without requiring additional sleep time. In particular, an increase in NREM stage 3 (DTS) during the sleep cycle has been reported as an important rest and recovery stage for the human body [44,45]. These findings indicate that VR-based meditation can effectively improve sleep, and the application of VR-based meditation can potentially improve an individual’s sleep quality in a short period, according to their convenience.

Based on the results of this study, the experimental group’s subjective and objective stresses decreased. Additionally, the heart rate variability (HRV) index was 8.65, indicating that the autonomic nervous system was stabilized after the study intervention [46]. In this way, meditation intervention using virtual reality had a positive effect on the subject’s emotions and thinking control during the stress resistance stage of general adaptation syndrome in Hans Selye [47], enabling him to cope with initial stress. In addition, it is believed that meditation intervention using virtual reality to relieve stress can increase an individual’s self-management capacity to reduce stress by increasing immersion and concentration more easily than the method used in existing meditation, thereby easily inducing mental and physical stability [48,49]. This effect is similar to a previous study that applied virtual reality nature meditation to college students [31]. The virtual reality-based meditation intervention had a positive effect in that it encouraged and supported the health care behaviors desired by each subject by reducing temporal and spatial constraints and providing access to various natural spaces.

In addition, a virtual reality-based meditation intervention that provides immersive images of real nature is believed to induce psychological stability and have a positive effect on stabilizing heart rate variability, an indicator of autonomic nervous system balance. These effects are similar to a previous study that applied mindfulness meditation to college students [50] and a previous study that applied short-term meditation training to healthy middle-aged adults [51]. According to research results, if the stabilization of the autonomic nervous system is maintained, it has a positive effect on stress management, maintenance of stable heart rate and blood pressure, improvement of sleep quality, body rest, and recovery from fatigue, and improves the ability to cope with stressors [17,52]. Accordingly, virtual reality-based meditation effectively regulates emotions and cognition in the early stages of stress by further increasing concentration and immersion in the program. In addition, it is believed that it potentially increases an individual’s ability to self-manage stress by more easily inducing mental and physical stability [48].

In this study, virtual reality-based meditation showed significant effects on improving sleep quality, reducing stress, and stabilizing the autonomic nervous system in nursing students. Therefore, the nursing significance of this study lies in the fact that applying a virtual reality-based meditation program is an effective primary prevention intervention for improving the health management lifestyle habits of early adult patients. The limitation of this study is that it was conducted on nursing students and cannot be applied to various population groups. Additional research is needed to apply the results to various population groups when conducting future studies. In addition, in order to verify the effectiveness of this study, it will be necessary to expand the scope of application to hospital clinical environments that have many time and space constraints, low sleep quality, and high stress.

## 5. Conclusions

Although this study’s intervention could not be expanded to participants in various fields, it was confirmed to be effective in improving sleep quality, reducing stress, and stabilizing heart rate variability by applying it to nursing students in their early adulthood. Based on the found results, virtual reality-based meditation, which provides immersive real-life natural images, is expected to be useful as an intervention for sleep hygiene and stress control.

## Figures and Tables

**Figure 1 healthcare-12-01581-f001:**
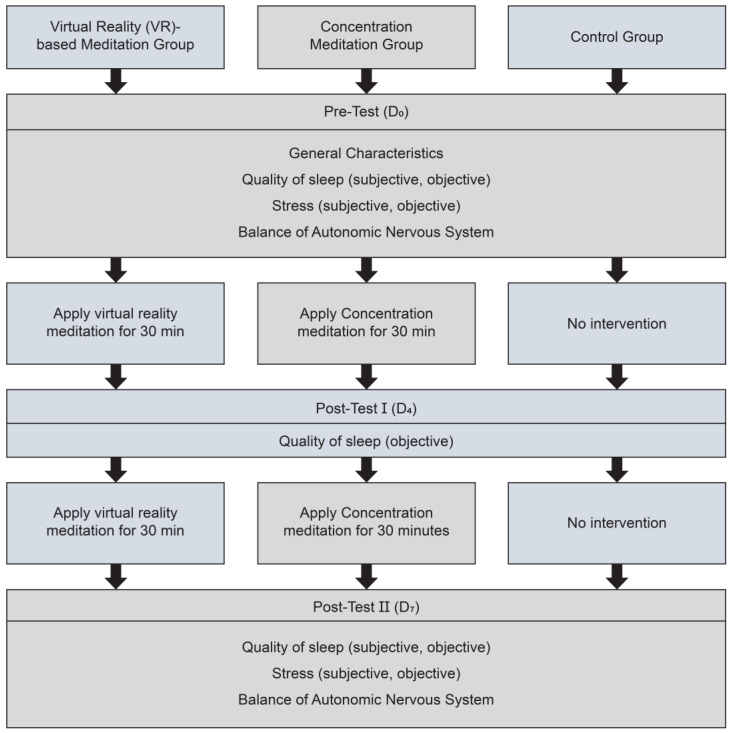
Design of this study.

**Figure 2 healthcare-12-01581-f002:**
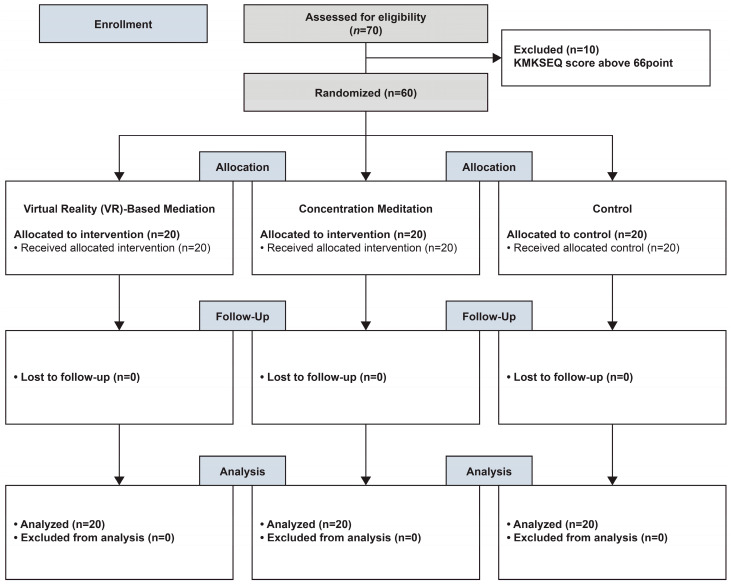
Flow diagram of the research process.

**Table 1 healthcare-12-01581-t001:** Participant selection and exclusion criteria.

**Selection Criteria**
Individuals who understood the aims of the study and voluntarily consented to participate Male and female nursing students aged at least 19 years who can communicate (nursing students undergoing theoretical education) Individuals with a subjective sleep quality score ≤66 Individuals with no visual problems Individuals with no auditory problems
**Exclusion Criteria**
Individuals on sleep-related medication Individuals currently diagnosed with and being treated for a physical or psychiatric disease Male and female nursing students aged at least 19 years who can communicate (nursing students in clinical practice)

**Table 2 healthcare-12-01581-t002:** Homogeneity test of general characteristics and dependent variable among three groups (n = 60).

Characteristic	Category	VRM. G (n = 20)	CM. G (n = 20)	Cont. G (n = 20)	χ^2^ or F	*p*
		Mean ± SD or N (%)	Mean ± SD or N (%)	Mean ± SD or N (%)		
Age (years)		21.6 ± 1.47	21.75 ± 1.48	22.35 ± 1.35	1.533	0.225
Sex	Male	5 (25%)	7 (35%)	7 (35%)	0.296	0.745
	Female	15 (75%)	13 (65%)	13 (65%)		
Grade	1 Grade	2 (10%)	0 (0%)	1 (5%)	1.868	0.164
	2 Grade	6 (30%)	6 (30%)	1 (5%)		
	3 Grade	11 (55%)	11 (55%)	16 (80%)		
	4 Grade	1 (5%)	3 (15%)	2 (10%)		
Sleep duration (h/day)	4–6	8 (40%)	8 (40%)	7 (35%)	0.060	0.942
	6–8	12 (60%)	11 (55%)	11 (55%)		
	over 8	0 (0%)	1 (5%)	2 (10%)		
Medication status	Yes	0 (0%)	0 (0%)	0 (0%)		
	No	20 (100%)	20 (100%)	20 (100%)		
Underlying disease	Yes	0 (0%)	0 (0%)	0 (0%)		
	No	20 (100%)	20 (100%)	20 (100%)		
Vital sign	SBP	114.65 ± 8.18	111.50 ± 7.96	115.70 ± 9.56	1.292	0.283
	DBP	69.15 ± 11.00	72.50 ± 8.31	75.80 ± 7.92	2.625	0.081
	Heart rate	80.70 ± 7.22	82.60 ± 5.52	79.70 ±6.02	1.096	0.341
KMLSEQ		46.60 ± 7.05	45.95 ± 8.86	47.70 ± 7.19	0.261	0.771
WASO (min)		51.25 ± 18.60	50.65 ± 8.28	51.95 ± 10.31	0.049	0.952
SE (%)		83.76 ± 5.53	83.97 ± 1.80	83.69 ± 2.03	0.035	0.966
DST (%)		13.95 ± 2.31	14.50 ± 1.88	14.25 ± 1.41	0.420	0.659
PSS		2.83 ± 0.69	2.91 ± 0.44	2.73 ± 0.39	0.600	0.552
SI		4.65 ± 1.53	4.35 ± 1.23	5.10 ± 1.17	1.643	0.203
HRV		7.66 ± 0.87	8.11 ± 1.31	8.28 ± 1.27	1.478	0.237

Note: VRM. G: virtual reality (VR)-based meditation group; CM. G: concentration meditation group; Cont. G: control group; mean ± SD: mean ± standard deviation; hr: hour; SBP: systolic blood pressure; DBP: diastolic blood pressure; KMLSEQ: Korean Modified Leeds Evaluation Questionnaire; WASO: wakefulness after sleep onset; SE: sleep efficiency; DST: deep sleep time; PSS: Perceived Stress Scale; SI: stress index; HRV: heart rate variability; min: minutes.

**Table 3 healthcare-12-01581-t003:** Comparison of the subjective quality of sleep among three groups (n = 60).

Variable		VRM. G (n = 20)	CM. G (n = 20)	Cont. G (n = 20)	F	*p*
		Mean ± SD	Mean ±SD	Mean ± SD		
KMLSEQ	D_0_	46.60 ± 7.05	45.95 ± 8.86	47.70 ± 7.19	0.261	0.771
	D_7_	61.35 ± 14.00	52.20 ± 9.32	45.45 ± 6.19	10.400	<0.001
	Difference (D_7_ − D_0_)	14.75 ± 14.57	6.25 ± 12.19	−2.25 ± 10.64	10.179	<0.001

Note: VRM. G: virtual reality (VR)-based meditation group; CM. G: concentration meditation group; Cont. G: control group; mean ± SD: mean ± standard deviation; KMLSEQ: Korean Modified Leeds Evaluation Questionnaire; D_0_: baseline; D_7_: last day of measurement.

**Table 4 healthcare-12-01581-t004:** Comparison of the objective quality of sleep among three groups (n = 60).

Variable		VRM. G (n = 20)	CM. G (n = 20)	Cont. G (n = 20)	F	*p*	F (*p*) †
		Mean ± SD	Mean ± SD	Mean ± SD			
WASO (min)	D_0_	51.25 ± 18.60	50.65 ± 8.28	51.95 ± 10.31	0.049	0.952	Time 0.120 (0.887) Group 8.106 (0.001) G × T 2.387 (0.055)
	D_4_	48.90 ± 15.16	50.20 ± 4.46	57.30 ± 7.97	3.916	0.025	
	D_7_	45.85 ± 6.37	48.60 ± 7.65	59.90 ± 7.93	20.540	<0.001	
	Difference (D_7_ − D_0_)	−5.40 ± 20.79	−2.05 ± 13.11	7.98 ± 14.15	3.599	0.034	
SE (%)	D_0_	83.76 ± 5.53	83.97 ± 1.80	83.69 ± 2.03	0.035	0.966	Time 0.571 (0.568) Group 15.553 (<0.001) G × T 3.123 (0.018)
	D_4_	84.46 ± 2.24	84.15 ± 1.39	81.86 ± 2.07	10.794	<0.001	
	D_7_	85.50 ± 1.44	84.70 ± 1.42	81.31 ± 1.86	39.108	<0.001	
	Difference (D_7_ − D_0_)	1.74 ± 5.88	0.73 ± 2.35	−2.37 ± 2.48	5.954	0.004	
DST (%)	D_0_	13.95 ± 2.31	14.50 ± 1.88	14.25 ± 1.41	0.420	0.659	Time 10.842 (<0.001) Group 11.308 (<0.001) G × T 4.458 (0.002)
	D_4_	16.40 ± 5.68	15.05 ± 1.82	13.95 ± 1.64	2.362	0.103	
	D_7_	19.65 ± 4.31	17.00 ± 4.04	14.00 ± 1.84	12.538	<0.001	
	Difference (D_7_ − D_0_)	5.70 ± 4.62	2.50 ± 4.59	−0.25 ± 2.69	10.698	<0.001	

Note: VRM. G: virtual reality (VR)-based meditation group; CM. G: concentration meditation group; Cont. G: control group; mean ± SD: mean ± standard deviation; †: repeated measures ANOVA; WASO: wakefulness after sleep onset; SE: sleep efficiency; DST: deep sleep time; D_0_: baseline; D_4_: the fourth day of research; D_7_: last day of measurement; min: minutes.

**Table 5 healthcare-12-01581-t005:** Comparison of subjective stress among three groups (n = 60).

Variable		VRM. G (n = 20)	CM. G (n = 20)	Cont. G (n = 20)	F	*p*
		Mean ± SD	Mean ± SD	Mean ± SD		
PSS	D_0_	2.83 ±0.69	2.91 ± 0.44	2.73 ± 0.39	0.600	0.552
	D_7_	2.24 ± 0.51	2.58 ± 0.45	2.69 ± 0.51	4.547	0.015
	Difference (D_7_ − D_0_)	−0.59 ± 0.76	−0.33 ± 0.56	−0.04 ± 0.61	3.552	0.035
SI	D_0_	4.65 ± 1.53	4.35 ± 1.23	5.10 ± 1.17	1.643	0.203
	D_7_	2.05 ± 0.75	3.50 ± 0.61	4.80 ± 1.20	47.784	<0.001
	Difference (D_7_ − D_0_)	−2.60 ±1.60	−0.85 ± 1.42	−0.30 ± 1.98	10.179	<0.001
HRV	D_0_	7.66 ± 0.87	8.11 ± 1.32	8.28 ± 1.27	1.478	0.237
	D_7_	8.65 ± 0.95	8.45 ± 1.15	7.75 ± 1.36	3.301	0.044
	Difference (D_7_ − D_0_)	0.99 ± 0.69	0.35 ± 0.93	−0.53 ± 1.24	12.126	<0.001

Note: VRM. G: virtual reality (VR)-based meditation group; CM. G: concentration meditation group; Cont. G: control group; mean ± SD: mean ± standard deviation; PSS: Perceived Stress Scale; SI: stress index; HRV: heart rate variability; D_0_: baseline; D_7_: last day of measurement.

## Data Availability

No new data were created or analyzed in this study. Data sharing is not applicable to this study.

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
