# Peer review of "Effects of Virtual Reality (VR)-Based Meditation on Sleep Quality, Stress, and Autonomic Nervous System Balance in Nursing Students"

_healthcare, 2024, doi:10.3390/healthcare12161581_

Round 1

Reviewer 1 Report

Comments and Suggestions for Authors

The authors conducted a very interesting and novel study, by illustrating the potential benefit of virtual reality (VR)-based meditation on sleep quality, stress, and heart rate (HR) variability. This is a still poorly studied topic with promising fields of application. As reported by the authors, after a VR-based meditation intervention, participants reported better sleep quality and reduced stress, with also a possible positive effect on HR variability.

Overall, the article is well-written and clear. However, I have a few comments.

1.     First of all, in the Introduction I would better describe the procedure and the results of the cited studies that used VR intervention. In addition, consider to cite this study: https://doi.org/10.1177/20552076221107887.

2.     In line 60 the authors state “no interventional study has examined how meditation affects sleep etc.”. I am not sure if this sentence is correct. Although there is still little evidence, other studies have also investigated the effect of meditation on these factors.

3.     Considering the limited evidence in the literature, the use of direct hypotheses should be avoided. In any case, the authors should explain why they assumed a difference between the groups. Rather, I would adopt a more cautious approach.

4.     Please justify the use of a Fitbit device for sleep monitoring, instead of the gold standard actigraphy/PSG.

5.     I believe that the experimental procedure description could be improved by removing the lists and adopting a more narrative approach.

6.     I would better explain the baseline assessment. It is not quite clear to me how long it lasted (I assume 1 day) and at what time of the day it was conducted.

7.     In the result tables the significant p-values should be in bold.

8. In the Discussion, the authors stated that after the intervention, HR variability met the ideal standards for ANS. I am not convinced by this statement. According to Sammito and Böckelmann (DOI: 10.1016/j.hrthm.2016.12.015), HR variability values > 80 in healthy young people are quite high. In addition, after the meditation, participants reported increased HR variability. How do you explain this?

9. I think there is a typo in line 355.

10. I detected some limitations that should be reported. a) the use of a between-group design, instead of a within-participant design; b) the different duration of the baseline assessment (1 day) and the intervention (7 days); c)  the meditation intervention is quite limited, based solely on breathing control; d) the absence of a real therapist for meditation training.

Comments on the Quality of English Language

English is quite good with few typos.

Author Response

I have uploaded it in the attached file below. Thank you for your good opinion. Please review it.

Reviewer 2 Report

Comments and Suggestions for Authors

Dear Authors,

Some concerns in relation to the study,

- The RCT should be registered in a trial register, and provide the number of the registartion.

- The study should follow the CONSORT guidelines, including de flow diagram.

- There is no dropout of participants in the study, how was the follow up?

- The intervention was for 7 days?

- Were there clinically relevant changes?

- Were there any limitations in the study?

- The conclusion at the end of the discussion should be remove to the conclusion section.

- References need to be reviewed and completed.

Thank you for the research.

Author Response

(The authors gave the same response as above.)

Reviewer 3 Report

Comments and Suggestions for Authors

Thank you for the possibility to review this manuscript. I appreciate the hard work and the high quality of this manuscript, congratulations! However, I need to point out some remarks to improve the quality of this manuscript:

1. Please indicate, whether your study was following the CONSORT guidelines. 

2. Please double-check the references as items 14, 20, 24, 38, and 39 are miswritten. I also consider it unnecessary to underline DOI in some references.

3. In subsection 2.2.1. Participant Selection and in section 2.3. Allocation of Participants, information regarding study participants allocation to groups is repeated. I suggest removing this information from section 2.3. 

4. Additionally, I suggest changing the name of subsection 2.3 to "Randomization and blinding". Providing information about blinding process can significantly increase the value of the study.

5. I suggest to rewrite the Results section as in description you repeat almost all information you placed in the tables (mainly Table 3). Also, please double-check the information provided in the description and the tables, as there is a mistake regarding SD value in the description of the Table 3 (line: 266), as well as in the description of Table 4 (line: 277).

6. Please correct the spaces whenever you indicate p value (e.g. "p<" or "p <") and the list of references (e.g. 12 and 17).

7. You can also remove the values and numbers in the Discussion section to make it more readable. 

Comments on the Quality of English Language

Please double-check for the spelling, puncttuation and grammar mistakes.

Author Response

(The authors gave the same response as above.)

Round 2

Reviewer 2 Report

Comments and Suggestions for Authors

Dear authors,

thank you for this new version.

In relation to this one, amd acording to the conclusion, limitations are part of the discusion. it sould not be in the conclusion section.

And what is the diference between the contents of the conclusion section and the last paragraph of the discusion that starts with "In conclusion,..."

Author Response

After review, our response to your feedback has been uploaded to the file below.
